# Fc-Modified Antibody in Hospitalized Severe COVID-19 Patients

**DOI:** 10.3390/vaccines13040372

**Published:** 2025-03-31

**Authors:** Felipe Dal-Pizzol, Suzana Margareth Lobo, Christopher Lucasti, Adam Abdul Hakeem Baidoo, Huo Su, Zhanghua Lan, Liangzhi Xie

**Affiliations:** 1Intensive Care Unit, Hospital São José, Criciúma 88806-000, SC, Brazil; fdpizzol@gmail.com; 2Laboratory of Experimental Pathophysiology, Graduate Program in Health Sciences, Universidade do Extremo Sul Catarinense, Criciúma 88806-000, SC, Brazil; 3Intensive Care Division, Hospital de Base, FAMERP, São José do Rio Preto 15090-000, SP, Brazil; suzanaalobo@gmail.com; 4South Jersey Infectious Disease, 730 Shore Road, Somers Point 08244, NJ, USA; infect123@aol.com; 5Beijing Engineering Research Center of Protein and Antibody, Sinocelltech Ltd., Beijing 100176, China; baidooadam@sinocelltech.com (A.A.H.B.); huo_su@sinocelltech.com (H.S.); zhanghua_lan@sinocelltech.com (Z.L.); 6Cell Culture Engineering Center, Chinese Academy of Medical Sciences & Peking Union Medical College, Beijing 100005, China

**Keywords:** SARS-CoV-2, Fc-modified antibody, severe COVID-19, mortality, hospitalization, antibody-dependent enhancement

## Abstract

**Background:** Hospitalized patients with severe COVID-19 are at high risk of clinical deterioration. **Methods:** A global, randomized, double-blinded, and placebo-controlled phase II trial that investigated the clinical efficacy of SCTA01, an Fc-modified monoclonal antibody, in patients hospitalized with severe COVID-19 during the Delta variant wave was performed. The primary outcome was time to clinical improvement up to Day 29. Secondary outcomes measured the all-cause mortality rate up to Day 29, time to SARS-CoV-2 RNA negativity up to Day 29, and the number of antibody-dependent enhancements. **Results:** From 27 March 2021, to 11 February 2022, 102 hospitalized adults with severe COVID-19 received a single intravenous infusion of SCTA01 15 mg/kg or 50 mg/kg or placebo in a 1:1:1 ratio. The median time to clinical improvement in the SCTA01 group was numerically shorter than that in the placebo group; however, the between group difference was statistically non-significant (SCTA01 15 mg/kg vs. placebo, HR 0.99, 95% CI 0.55–1.77, *p* = 0.742; SCTA01 50 mg/kg vs. placebo, HR 1.07, 95% CI 0.61–1.88, *p* = 0.095). The median time to achieve a negative SARS-CoV-2 status was shorter in the SCTA01 15 mg/kg group (14.0 days vs. 27.0 days) but not in the SCTA01 50 mg/kg group (28.0 days vs. 27.0 days) compared to the placebo group. Adverse events were comparable across all groups, and no treatment-related serious adverse event or antibody-dependent enhancement was reported. **Conclusions:** The Fc-modified antibody was safe but lacked significant clinical efficacy in vivo, likely due to the SARS-CoV-2 viral mutation.

## 1. Introduction

SARS-CoV-2 infection generally leads to mild or moderate disease; however, earlier in the pandemic, approximately 10–30% of infected individuals developed severe disease with detrimental complications [1,2,3]. Monoclonal antibodies are known to provide immediate, passive immunity to patients with acute infections and sometimes late-stage disease due to their high specificity and ability to boost immune responses [4]. Although antibody use in severe COVID-19 patients suggested that recipients had slightly improved outcomes compared to those who did not receive it, their efficacy waned as the disease progressed and new variants emerged.

Severe COVID-19 hospitalizations increased dramatically during the Delta wave period. From March 2020 to March 2021, over 90% of adults hospitalized with severe COVID-19 had at least one underlying medical condition, with the most prevalent conditions being hypertension and disorders of lipid metabolism. Conversely, obesity, diabetes with complications, and anxiety disorders were found to be the most critical risk factors for severe COVID-19 illness [5]. Severe COVID-19 triggers a wide range of immune interactions (innate and adaptive responses), which leads to the overproduction of pro-inflammatory cytokines and disrupts normal immune responses [6]. It is characterized by a dysregulated response to the virus, often leading to uncontrolled viral replication in the pathogenesis of this disease [7].

SCTA01 is a humanized monoclonal IgG1 antibody with an LALA-modified Fc region that binds to the SARS-CoV-2 spike protein, thereby effectively blocking its interaction with ACE2 receptors [8]. This antibody’s Fc segment has been modified to reduce interactions with human Fc receptors and complement, which could potentially mitigate hyper-inflammation and antibody-dependent enhancement (ADE) in severely ill COVID-19 patients.

This paper reports the phase II findings from a global, multi-regional study of SCTA01 conducted in hospitalized severe COVID-19 patients, which was terminated due to the emergence of the Omicron variant.

## 2. Methods

### 2.1. Study Design

SCTA01-B301 was a phase II/III randomized, global, double-blinded, placebo-controlled study evaluating SCTA01 (15 mg/kg and 50 mg/kg) in hospitalized severe COVID-19 patients. A total of 285 patients were to be randomized in a 1:1:1 ratio among the treatment and placebo groups, with 95 patients in each group. An interim analysis was planned after monitoring the 285th patient for time to clinical improvement (TTCI) up to Day 29. An Independent Data Monitoring Committee (IDMC) supervised the safety data, with all necessary approvals obtained per Good Clinical Practice. Patients or their representatives provided informed consent before participation. Trial registration: ClinicalTrials.gov Identifier NCT04644185 (22 November 2020).

### 2.2. Trial Participants

Eligible patients were adults ≥ 18 years old hospitalized with severe COVID-19, confirmed by a polymerase chain reaction (PCR) test, and ≤14 days from symptoms onset. Severe COVID-19 was defined according to the US National Institute of Health (NIH) criteria (at least one): (1) respiratory rate ≥ 30 breaths per minute; (2) pulse oxygen saturation (SpO_2_) ≤ 93% on room air at sea level; (3) arterial partial pressure of oxygen (PaO_2_)/fraction of inspiration oxygen (FiO_2_) < 300 mmHg or SpO_2_/FiO_2_ ≤ 315 mmHg; (4) lung infiltrates > 50%. Patients with critical COVID-19 were excluded. According to the NIH definition, critically ill patients are those having one of the following conditions: (1) respiratory failure and need invasive mechanical ventilation; (2) septic shock; (3) multiple organ dysfunction (detailed eligibility criteria are provided in the protocol in Appendix A). Patients received a single intravenous infusion of SCTA01 or placebo on Day 1, along with the best supportive care. Patients were monitored daily until Day 29, with follow-up on Days 60 and 120.

### 2.3. Randomization and Blinding

Patients were randomly assigned to receive SCTA01 at either 15 mg/kg, 50 mg/kg, or placebo in a 1:1:1 ratio, with the randomization schedule stratified by country. To ensure unbiased treatment assignment and concealment of treatment allocation, randomization data were kept strictly confidential until final analyses. They were accessible only to the IDMC team and the unblinded statistician. Additionally, the treatment identity was concealed using identical packaging, labeling, schedule of administration, and appearances for SCTA01 and the matching placebo.

### 2.4. Efficacy and Safety Assessments

The primary endpoint was time to clinical improvement up to Day 29, assessed by an 8-point ordinal scale. TTCI was defined as the time (in days) from randomization to the first day on which a patient satisfies points 1, 2, or 3 on the 8-point ordinal scale and maintains a score ≤3 for at least 48 h (initial improvement) and then maintains this score up to Day 29. Patients were discharged from acute care once their baseline vital signs had returned to normal, without requiring supplemental oxygen, and after consecutive negative PCR test results. Secondary outcomes assessed by clinical efficacy included the all-cause mortality rate; virologic efficacy was assessed by the time to SARS-CoV-2 RNA negativity from baseline. Safety assessments included the cumulative incidence of serious adverse events (SAEs), the incidence of Grade 1, 2, 3, and 4 adverse events (AEs), and the proportion of patients suspected of experiencing antibody-dependent enhancement (ADE).

### 2.5. Statistical Analysis

The primary analysis included all randomized patients who received the study drug. Patients were analyzed according to their assigned treatment. Patients lost to follow-up or terminated early before the primary endpoint were censored at their last assessment. Those who completed follow-up without meeting the endpoint were censored on Day 29. Patients who took prohibited medications before Day 29 were treated as treatment failures and censored at that time.

The primary analysis utilized the stratified log-rank test to compare the time to clinical improvement between the treatment and control groups through Day 29, accounting for factors such as country, duration of symptoms, comorbidity, and supportive treatment. Small strata were combined for statistical viability. The Kaplan–Meier estimate of TTCI survival was calculated for each group, and the hazard ratio for SCTA01 versus placebo was determined using a stratified Cox proportional hazards model. Data on TTCI events and censored patients were summarized by treatment group, along with the percentage of patients on an 8-point ordinal scale up to Day 29. If proportional hazard assumptions were not met, the Cochran–Mantel–Haenszel test was used for comparison.

The secondary endpoint, the all-cause mortality rate, was analyzed using a logistic regression model. The odds ratio (OR) and its two-sided 95% confidence interval for the treatment effect were provided. The time to SARS-CoV-2 RNA negativity was summarized through Kaplan–Meier curves and 95% confidence bounds. A stratified Cox model was used to estimate the hazard ratio and two-sided 95% confidence intervals. Safety summaries included the number and percentage of patients presented, along with a breakdown by system organ class (SOC) and preferred term (PT), each being counted only once at each summarization level. Full details are provided in Appendix A.

The safety summaries included all patients from the SCTA01 and placebo groups from the safety set, with adverse events (AEs) tabulated by treatment. The number and percentage of patients with at least one AE were presented, along with a breakdown by system organ class (SOC) and preferred term (PT). Each patient was counted once at each summarization level. The statistical analysis plan (SAP) was finalized before the clinical database was locked, and two-sided *p*-values at a 5% significance level were provided for interpretation. For more details, refer to the Statistical Analysis Plan (SAP).

### 2.6. Early Study Termination

The project team maintained close oversight, accompanied by frequent IDMC reviews of the safety data. This study was terminated early on 29 December 2021, following the emergence of the Omicron variant, after enrolling approximately 35% of the predefined sample size for phase II (295 patients). Investigators were directed to conduct early termination visits for patients who had not completed the study. Ethical committees and regulatory authorities were informed of the early termination, with the last visit on 11 February 2022.

## 3. Results

### 3.1. Trial Participants

This study was from 27 March to 29 December 2021, and 116 patients were screened across 11 investigational centers in five countries: the United States of America, Argentina, Brazil, Chile, and Peru. Finally, 103 hospitalized patients with severe COVID-19 were randomly assigned in a 1:1:1 ratio to receive SCTA01 15 mg/kg (*n* = 33), SCTA01 50 mg/kg (*n* = 34), or a placebo (*n* = 35). Among these patients, 56 (54.9%) were seropositive, 36 (35.3%) were seronegative, and the serostatus was unknown for 10 (9.8%) at baseline. All patients received the best supportive treatment, including corticosteroids (primarily dexamethasone), remdesivir, or other COVID-19 drug therapy. The analysis population consisted of only 102 patients, as one patient withdrew (Figure 1).

### 3.2. Baseline Demographic and Clinical Characteristics

One hundred two patients were included in the intention-to-treat (ITT) population, comprising sixty-eight males and a median age of 48. The median duration from COVID-19 symptom onset to enrollment was 9 days in the treatment groups and 7 days in the placebo group. Presenting comorbid conditions were similar between groups. Over two-thirds of patients had at least one co-existing illness, with hypertension being the most common. Approximately 89.2% (91/102) of patients completed the Day 29 follow-up visit, which served as the primary endpoint. The remaining patients were in follow-up until the study’s termination. Baseline demographics and clinical characteristics were balanced between groups (Table 1).

### 3.3. Efficacy Outcomes

The primary objective was to compare the effect of SCTA01 to placebo on patient survival, with time to clinical improvement up to Day 29 as the primary outcome. Key secondary endpoints included the all-cause mortality rate up to Day 29 and the time to SARS-CoV-2 RNA negativity.

### 3.4. Time to Clinical Improvement

The median time to clinical improvement for all patients was numerically shorter but statistically non-significant, at 9.0 days for both SCTA01 groups and 10 days for the placebo (Table 2). No significant difference in time to clinical improvement was observed in the treatment groups and placebo: SCTA01 15 mg/kg vs. placebo, HR 0.99, 95% CI 0.55–1.77, *p* = 0.742; SCTA01 50 mg/kg vs. placebo, HR 1.07, 95% CI 0.61–1.88, *p* = 0.095 (Figure 2A). The cumulative incidence of patients who achieved score 1 (not hospitalized) on Day 29 was 26 (78.8%), 25 (73.5%), and 24 (68.6%) in SCTA01 15 mg/kg, SCTA01 50 mg/kg, and placebo, respectively. The entire distribution of patients on the 8-point scale, for which complete data are available for all patients, is shown in (Appendix A).

### 3.5. All-Cause Mortality Rate

The all-cause mortality rate up to Day 29 was numerically lower in the SCTA01 treatment groups (15 mg/kg: (6.1%), 50 mg/kg: (5.9%)) than in the placebo group (8.6%); however, it was with no statistical significance (Table 2). Throughout the study duration, there were nine recorded deaths, with four related to COVID-19. Other causes of death included pulmonary sepsis, cardiac failure, respiratory failure, sepsis, and multiple organ dysfunction syndrome.

### 3.6. Time to SARS-CoV-2 RNA Negativity

There was a clear trend with SCTA01 15 mg/kg showing a lower median time to SARS-CoV-2 negativity (14.0 days) compared to the placebo group (27.0 days) (Figure 2B). Seropositive patients in the SCTA01 15 mg/kg group also exhibited a lower median time to SARS-CoV-2 RNA negativity compared to the SCTA01 50 mg/kg and placebo groups. Seronegative patients generally had a longer median time to SARS-CoV-2 negativity across all groups (Table 2).

### 3.7. Non-Medication Supportive Measures

The median duration of time for supplemental oxygen support for 1 day or longer was shorter in the treatment group than in the placebo group. A lower proportion of patients in the treatment groups required invasive mechanical ventilation or extracorporeal membrane oxygenation (ECMO) for 1 day or longer compared to those in the placebo group. However, the median time on invasive support was similar across all groups (Table 2).

### 3.8. Safety Outcomes

In the safety analysis of 102 randomized and dosed patients, 59 (57.84%) experienced at least one treatment-emergent adverse event (TEAE), most of which were mild or moderate. Only one patient in the SCTA01 15 mg/kg group experienced increased transaminases related to treatment, but this did not pose any threat to the subject’s safety (Table 3). This patient had elevated liver transaminases at baseline. Gastrointestinal disorders were the most common, with constipation being the leading TEAE. Serious adverse events (SAEs) occurred in 17 patients (16.7%), primarily respiratory, with respiratory failure being the most frequently reported. No patients withdrew due to study drug-related adverse events, and there were no severe allergic reactions, cytokine release, or adverse drug reactions reported. Full details of ≥5% adverse events and SAEs are presented in Appendix A and Appendix A, respectively.

## 4. Discussion

In this global trial involving hospitalized severe COVID-19 patients, the treatment groups showed numerically improved outcomes compared to the placebo group, although the difference was not statistically significant. The time to clinical improvement was similar for both SCTA01 15 mg/kg (HR 0.99, 95% CI 0.55–1.77, *p* = 0.742) and SCTA01 50 mg/kg (HR 1.07, 95% CI 0.61–1.88, *p* = 0.095). On Day 29, 78.8% of the SCTA01 15 mg/kg group, 73.5% of the SCTA01 50 mg/kg group, and 68.6% of the placebo group scored one on the ordinal scale (indicating no hospitalization and no activity limitations) (Table 2).

Although COVID-19 is primarily a mild-to-moderate disease, a subset of patients presented with high viral titers and developed severe symptoms, which led to organ failure or even death [9,10]. Patients in the low-dose SCTA01 treatment group showed a shorter median duration compared to placebo (14 days vs. 27 days) in becoming SARS-CoV-2 RNA negative, albeit statistically non-significant (SCTA01 15 mg/kg vs. placebo HR 1.57, 95% CI 0.91–2.74, *p* = 0.359; SCTA01 50 mg/kg vs. placebo HR 1.11, 95% CI 0.64–1.92, *p* = 0.883). A likely explanation for this observation could be the pharmacokinetics of the low-dose treatment triggering a more targeted immune response leading to quicker viral clearance, considering that the proportion of patients with high viral titers was similar across the three groups (SCTA01 15 mg/kg, 24 (72.7%), SCTA01 50 mg/kg, 25 (73.5%), and placebo 22 (62.9%)). A study by Lundgren et al. reported that high doses of neutralizing monoclonal antibodies have limited effectiveness among hospitalized COVID-19 patients [11].

Severe COVID-19 patients typically experience clinical deterioration due to the combined effects of viral load and inflammation [7,12,13,14]. Throughout the entire study duration, nine deaths (8.8%) were recorded, with four COVID-19-related (one patient received SCTA01 50 mg/kg, and three patients received placebo) and five non-COVID-19–related deaths (pulmonary sepsis, sepsis, cardiac failure, respiratory failure, and multiple organ dysfunction syndromes). Severe COVID-19 clinical deterioration usually happens in the second or third week of the illness, indicating a potentially evolving secondary immune response [10,15]. This could explain the high proportion of deaths prior to Day 29 (*n* = 7, with six occurring before Day 14) compared to post-Day 29 deaths (*n* = 2). The all-cause mortality rate up to Day 29 was similar in both treatment groups compared to the placebo (SCTA01 15 mg/kg: two (6.1%), SCTA01 50 mg/kg: two (5.9%), placebo: three (8.6%))—a trend consistent with other studies. A randomized trial of Tixagevimab–Cilgavimab in adults with prior symptoms up to 12 days and hospitalized for COVID-19 reported a numerically similar mortality rate in the treatment group (9%) compared to the placebo group (12%) [16]. Additionally, the RECOVERY trial of casirivimab and imdevimab in hospitalized COVID-19 patients reported a 28-day mortality rate of 19% in the treatment group, compared to 21% in the standard care group, among all patients regardless of baseline antibody status [17].

Evidence suggests that the activity of IgG antibodies in viral infections relies not only on neutralization [18,19] but also on the interactions of the Fc domains with Fcγ receptors, which can initiate effector mechanisms. While these interactions are important, they may also contribute to inflammation and ADE of disease, raising concerns for antibody therapies in the context of SARS-CoV-2 infection [20,21]. To mitigate the risk of ADE triggering harmful immunopathology [22], therapeutic monoclonal antibodies (mAbs) have a modified Fc region [23]. Specifically, the SCTA01 heavy chain was altered with an LALA mutation to reduce binding to Fc receptors and C1q complement, thereby lowering the potential for ADE. SCTA01 demonstrated protective neutralization against early SARS-CoV-2 strains in preclinical studies; however, reduced Fc engagement might lead to lower antiviral efficacy in vivo, as reflected in trial results [11,24].

Managing severe COVID-19 patients, particularly during the early stages of the pandemic, was challenging due to the complexity of their immunologic responses and comorbidities. Furthermore, SARS-CoV-2 exhibits a high mutation rate, particularly in the spike protein, which is the target of many monoclonal antibodies. These mutations altered viral proteins, effectively diminishing the effectiveness of therapeutic agents and neutralizing the action of antibodies [25]. Initially, guidelines recommended remdesivir and dexamethasone for mild to moderate cases; however, there was no consensus on the treatment for severe cases [26]. Corticosteroids, such as dexamethasone, may reduce inflammation and prevent respiratory failure; however, their effectiveness depends on the patient’s condition and the need for respiratory support. In our trial, nearly all patients received corticosteroids, primarily dexamethasone. The RECOVERY trial indicated that high doses of corticosteroids increased mortality in patients who did not require oxygen, while dexamethasone reduced 28-day mortality in patients who needed ventilation or oxygen [27]. On the other hand, dexamethasone reduced 28-day mortality in hospitalized COVID-19 patients who were on invasive mechanical ventilation or oxygen alone but not in those without respiratory support [28]. Our heterogeneous trial population required varying levels of respiratory support. A study of 24,856 adults hospitalized between 1 May 2020, and 3 May 2021, found that remdesivir treatment was linked to a reduced risk for inpatient mortality [29]. A Cochrane review suggested a minimal effect on overall or in-hospital mortality for moderate to severe cases [30]. Late administration of remdesivir was associated with more complications [31]. In our trial, 14% (*n* = 14) of those who received remdesivir had a median COVID-19 symptom onset 9 days before enrollment. Overall, the treatment group showed shorter supplemental oxygen durations and fewer patients requiring invasive ventilation compared to the placebo group.

No significant differences in adverse events were found between the treatment and placebo groups. The most common TEAEs were constipation and pyrexia. Most TEAEs were mild or moderate in severity. No deaths were attributed to the study treatment. Notably, no severe allergic reactions, cytokine releases, or ADE of COVID-19 were observed in this study.

The low-dose treatment showed a non-significant shorter time to SARS-CoV-2 negativity compared to the placebo, but no clinical benefit.

### Study Limitations

Our trial has several limitations. First, the small sample size prevented robust subgroup analyses. Second, this study primarily occurred before the widespread use of SARS-CoV-2 vaccines, making it challenging to extrapolate our results to vaccinated or boosted patients. Finally, the prevailing SARS-CoV-2 sub-lineage at the time (e.g., Delta variant) significantly constrains the generalizability of our results.

## 5. Conclusions

Although the Fc-modified antibody was safe in treating severe COVID-19, it did not show significant clinical efficacy.

## Figures and Tables

**Figure 1 vaccines-13-00372-f001:**
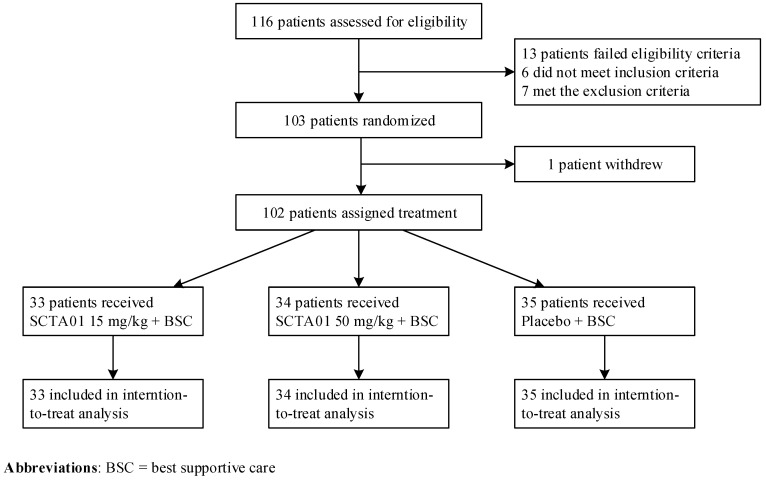
Disposition of trial patients. One hundred sixteen adults were screened for eligibility, and one hundred two were randomized to receive SCTA01 15 mg/kg, SCTA01 50 mg/kg, or placebo. One hundred two patients were included in the intention-to-treat (ITT) population for safety and efficacy analyses.

**Figure 2 vaccines-13-00372-f002:**
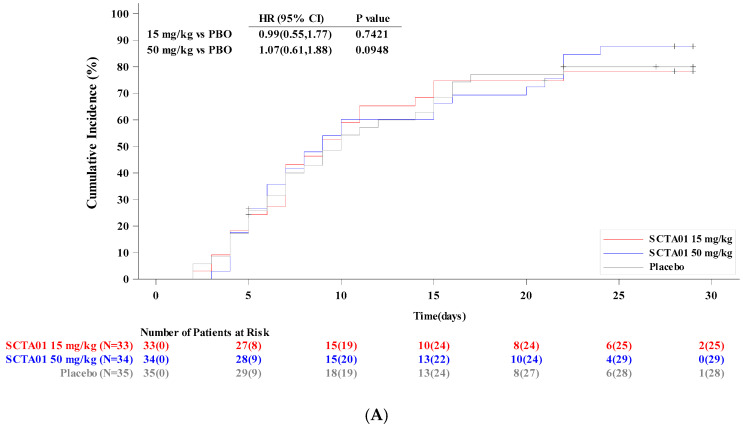
(**A**). Cumulative incidence of time to clinical improvement up to Day 29. (**B**). Cumulative incidence of time to SARS-CoV-2 negativity up to Day 29. **Abbreviations**: N = total number of patients; PBO = placebo; HR = hazard ratio; *p*-*value* = probability value; CI = confidence interval.

**Table 1 vaccines-13-00372-t001:** Baseline demographic and clinical characteristics.

	SCTA01 15 mg/kg N = 33, *n* (%)	SCTA01 50 mg/kg N = 34, *n* (%)	Placebo N = 35, *n* (%)
**Age in years,**			
18–49	19/33 (57.6)	18/34 (52.9)	21/35 (60.0)
≥50	14/33 (42.4)	16/34 (47.1)	14/35 (40.0)
Median (Min, Max)	48.0 (1, 81)	48.0 (30, 69)	44.0 (18, 76)
**Gender, *n* (%)**			
Male	21/33 (63.6)	22/34 (64.7)	25/35 (71.4)
Female;	12/33 (36.4)	12/34 (35.3)	10/35 (28.6)
**Race or ethnic group, *n* (%)**			
American Indian or Alaska Native	2/33 (6.1)	3/34 (8.8)	4/35 (11.4)
Asian	0/33	1/34 (2.9)	0/35
Black or African American	5/33 (15.2)	3/34 (8.8)	0/35
White	22/33 (66.7)	25/34 (73.5)	0/35
Hispanic or Latino	19/33 (57.6)	16/34 (47.1)	18/35 (51.4)
Not Reported or Unknown	4/33 (12.1)	2/34 (5.9)	2/35 (5.7)
**BMI (kg/m^2^)**			
<30	19/33 (57.6)	16/34 (47.1)	11/35 (31.4)
≥30	14/33 (42.4)	18/34 (52.9)	24/35 (68.6)
Mean (SD)	30.6 (7.53)	32.3 (8.44)	32.4 (5.53)
**Primary diagnosis, *n* (%)**			
Laboratory-confirmed SARS-CoV-2 infection	32/33 (97.0)	33/34 (97.1)	34/35 (97.1)
SpO_2_ ≤ 93% on room air a	31/33 (93.9)	31/34 (91.2)	34/35 (97.1)
PaO_2_/FiO_2_ < 300 mmHg	14/33 (42.4)	17/34 (50.0)	15/35 (42.9)
SpO_2_/FiO_2_ ≤ 315 mmHg	11/33 (33.3)	14/34 (41.2)	13/35 (37.1)
Lung infiltrates > 50%	15/33 (45.5)	15/34 (44.1)	14/35 (40.0)
Respiratory rate > 30 breaths/min	6/33 (18.2)	9/34 (26.5)	4/35 (11.4)
**Clinical status (8-point scale), *n* (%)**			
Score 5	32/33 (97.0)	33/34 (97.1)	32/35 (91.4)
Score 6	1/33 (3.0)	1/34 (2.9)	3/35 (8.6)
**Co-existing illness, *n* (%)**			
Any	24/33 (72.7)	23/34 (67.6)	24/35 (68.6)
Hypertension	12/33 (36.4)	10/34 (29.4)	11/35 (31.4)
Diabetes	7/33 (21.2)	4/34 (11.8)	2/35 (5.7)
Obesity	6/33 (18.2)	9/34 (26.5)	6/35 (17.1)
Chronic kidney disease	1/33 (3.0)	1/34 (2.9)	0/35
Generalized anxiety disorders	4/33 (12.1)	1/34 (2.9)	3/35 (8.6)
**Duration of symptoms before enrollment in days, *n* (%)**			
1–7	5/33 (15.2)	11/34 (32.4)	19/35 (54.3)
8–14	28/33 (84.8)	23/34 (67.6)	16/35 (45.7)
Median (Min, Max)	9.0 (1, 11)	9.0 (1, 11)	7.0 (1, 14)
**Baseline serostatus, *n* (%)**			
Seropositive	17/33 (51.5)	21/34 (61.8)	18/35 (51.4)
Seronegative	12/33 (36.4)	10/34 (29.4)	14/35 (40.0)
Unknown	4/33 (12.1)	3/34 (8.8)	3/35 (8.6)
**Medication use, *n* (%)**			
Remdesivir	4/33 (12.1)	4/34 (11.8)	6/35 (17.1)
Dexamethasone	32/33 (97.0)	33/34 (97.1)	35/35 (100)

**Abbreviations**: *n* = number of patients; N = total number of patients; BMI (body mass index) = Weight (kg)/Height (m)^2^; SD = standard deviation; Min = minimum; Max = maximum.

**Table 2 vaccines-13-00372-t002:** Study efficacy outcomes.

	SCTA01 15 mg/kg (N = 33)	SCTA01 50 mg/kg (N = 34)	Placebo (N = 35)
**Primary outcome**			
Time to clinical improvement, median (days)	9.0 (7.0, 14.0)	9.0 (6.0, 15.0)	10.0 (7.0, 15.0)
**Secondary outcomes**			
Mortality rate **up to Day 29**, n (%)	2/33 (6.1)	2/34 (5.9)	3/35 (8.6)
Time to SARS-CoV-2 negativity for all patients, median (days)	14.0 (8.0, 29.0)	28.0 (9.0, 31.0)	27.0 (11.0, 32.0)
*** Time to SARS-CoV-2 negativity in seronegative patients, median (days)* *(total n = 36)*	29.0 (3, 123)	116.0 (5, 118)	29.0 (16, 124)
*** Time to SARS-CoV-2 negativity in seropositive patients, median (days)* *(total n = 56)*	13 (8, 16)	29 (5, 31)	15 (8, 46)
**Non-medication measures** **(≥1 day)**			
Supplemental oxygen, n (%) Median (Min, Max) (days)	8 (24.2) 6.5 (2, 10)	11 (32.4) 3.0 (2, 16)	10 (28.6) 9.0 (4, 21)
Non-invasive ventilation, n (%) Median (Min, Max) (days)	5 (15.2) 6.0 (2, 9)	1 (2.9) 8.0 (8, 8)	3 (8.6) 4.0 (1, 4)
Invasive ventilation/ECMO, n (%) Median (Min, Max) (days)	4 (12.1) 13 (3, 29)	5 (14.7) 13 (1, 28)	9 (25.7) 10 (3, 23)

**Abbreviations:***n* = total number of patients in the seropositive and seronegative; N = total number of patients in the treatment group; ECMO = extracorporeal membrane oxygenation. ******** *Subgroup analysis based on baseline serostatus*.

**Table 3 vaccines-13-00372-t003:** Summary of treatment-emergent adverse events.

	SCTA01 15 mg/kg N = 33, (%)	SCTA01 50 mg/kg N = 34, (%)	Placebo N = 35, (%)
Any TEAEs	18/33 (54.5)	21/34 (61.8)	20/35 (57.1)
Related to the study treatment	1/33 (3.0)	0/34	0/35
TEAEs by maximum severity			
Grade 1	7/33 (21.2)	13/34 (38.2)	9/35 (25.7)
Grade 2	4/33 (12.1)	2/34 (5.9)	6/35 (17.1)
Grade 3	3/33 (9.1)	0/34	3/35 (8.6)
Grade 4	1/33 (3.0)	4/34 (11.8)	2/35 (5.7)
Grade 5	3/33 (9.1)	2/34 (5.9)	0/35
Any Grade 3 or above TEAEs	7/33 (21.2)	6/34 (17.6)	5/35 (14.3)
Related to study treatment	1/33 (3.0)	0/34	0/35
Any TEAEs leading to study treatment discontinuation	1/33 (3.0)	0/34	0/35
Any TEAEs leading to study discontinuation	2/33 (6.1)	1/34 (2.9)	0/35
Any allergic acute reactions	0/33	0/34	0/35
Any Grade 3 or above serious TEAEs	7/33 (21.2)	5/34 (14.7)	5/35 (14.3)

**Abbreviations:** N = total number of patients; TEAEs = treatment-emergent adverse events; SAE = serious adverse event.

## Data Availability

The data for this study are provided in the paper or Appendix A. The protocol and statistical analysis plan are provided as Appendix A. The trial has been registered on ClinicalTrials.gov under the identifier NCT04644185.

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
