# Peer review of "Fc-Modified Antibody in Hospitalized Severe COVID-19 Patients"

_vaccines, 2025, doi:10.3390/vaccines13040372_

Round 1

Reviewer 1 Report

Comments and Suggestions for Authors

This paper describes a clinical trial result of a therapeutic mAb on severe COVID-19 before the vaccine is available. Although the result was not statistically significant, it does present an authentic data of the real world experiment result and may still be informative. The only major concern of mine is that after COVID-19 vaccine is available, the significance of mAb treatment would be greatly reduced, and especially in year 2025, when Omicron and their descendents are prevalent, these circulating strains are no longer neutralized by SCTA01. 

There are some minor issues concerning the formatting and number of digits, which I included in the attachment. But they don't affect this paper's conclusion.

Comments on the Quality of English Language

Some of the wordings can be improved. (See attachment)

Author Response

Dear Reviewer

We appreciate the opportunity to submit the revised manuscript “Fc-Modified Antibody in Hospitalized Severe COVID-19 Patients” for publication in Vaccines.

We have revised the manuscript in response to the feedback and suggestions provided. The changes have been highlighted within the manuscript. Our responses to the comments, along with the corresponding modifications is in the attachment below. All authors have approved the revised manuscript. Thank you for your time and effort.

Reviewer 2 Report

Comments and Suggestions for Authors

Filipe Dal-Pizzol and colleagues present in the manuscript “Fc-Modified Antibody in Hospitalized Severe COVID-19 Patients” a phase II report on the clinical efficacy of SCTA01 monoclonal antibody in the treatment of severe COVID-19 patients. SCTA01 is a monoclonal antibody of the IgG1 subtype, primarily targeting the receptor-binding domain (RBD) of the SARS-CoV-2 spike protein. This Ab has been developed to have an Fc-mutated (LALA) modification that reduces antibody-dependent enhancement (ADE) and antibody-dependent cell cytotoxicity (ADCC) ability without impairing its high-affinity neutralizing ability. The study has not the statistical power to fully demonstrate the benefits of getting SCT01 during COVID infection and thus represents a negative/non conclusive data report.

In its category the report is well written and objectives clearly stated, with the changes suggested below the manuscript can be consider for publication.

Minor comments:

The study has been, initially, designed to enrol a higher number of patients than the one the authors effectively describe. In fact, the numbers do not fit throughout the methods. I suggest that, already in the first section, authors state the causes why the study has been shortened, or as alternative, to put a small note that leads to the last point of the methods where the justification for the shortening is given.

Give the rational or a reference for the choice on the doses used for therapy.

Figure legend 2: Please provide a title to the figure and describe in more detail both the graphs and tables below each plot. Abbreviations, that now are in duplicate, may be compacted at a unique paragraph at the end of the figure legend.

Author Response

Dear Reviewer

We appreciate the opportunity to submit the revised manuscript “Fc-Modified Antibody in Hospitalized Severe COVID-19 Patients” for publication in Vaccines.

We have revised the manuscript in response to the feedback and suggestions provided. The changes have been highlighted within the manuscript. Our responses to the comments, along with the corresponding modifications, are in the attached file below. All authors have approved the revised manuscript. Thank you for your time and effort.
